# Plastic Recycling Practices in Vietnam and Related Hazards for Health and the Environment

**DOI:** 10.3390/ijerph18084203

**Published:** 2021-04-15

**Authors:** Stefan Salhofer, Aleksander Jandric, Souphaphone Soudachanh, Thinh Le Xuan, Trinh Dinh Tran

**Affiliations:** 1Institute of Waste Management, University of Natural Resources and Life Sciences, 1190 Vienna, Austria; Aleksander.jandric@boku.ac.at (A.J.); Souphaphone.soudachan@boku.ac.at (S.S.); 2Vietnam Cleaner Production Centre Co. Ltd, Hanoi 11413, Vietnam; thinh.lx@vncpc.org; 3School of Environmental Science and Technology, Hanoi University of Science and Technology, Hanoi 11615, Vietnam; 4Faculty of Chemistry, University of Science, Vietnam National University, No. 19 Le Thanh Tong Street, Hanoi 11021, Vietnam; trinhtd@vnu.edu.vn

**Keywords:** plastic recycling, informal recycling, health risk, environmental impacts, craft village, Vietnam

## Abstract

Waste plastic today is a global threat. The rapid increase in global production and use has led to increasing quantities of plastics in industrial and municipal waste streams. While in industrialized countries plastic waste is taken up by a waste management system and at least partly recycled, in low-income countries adequate infrastructure to collect and treat waste adequately is often not in place. This paper analyzes how plastic waste is handled in Vietnam, a country with a fast-growing industry and growing consumption. The recycling of plastic waste typically takes place in an informal context. To demonstrate this in more detail, two rural settlements—so-called craft villages—are taken as case studies. Technologies and processes for plastic recycling are described and related risks for human health and the environment are shown, as well as the potential for the improvement of this situation.

## 1. Introduction

Plastic has become one of the most important materials used in varied industries due to its versatile properties and low cost. Plastic production and consumption have been increasing significantly since the 1950s [1]. About 7800 million tonnes (Mt) of plastic resins and fiber have been manufactured during the period of 1950 up to 2015. The global annual plastic production has been scaling up from 2 Mt in 1950 to 381 Mt in 2015. The trend of plastic production is growing at a fast rate, and it is estimated that it may reach up to 34,000 Mt of the total plastic ever manufactured by the year 2050 [2].

In parallel to plastic production, global plastic waste has also had a significant increase. The share of plastic in municipal waste has remarkably increased from less than 1% in 1960 to 10% in 2005. It was estimated that only 9% of the total plastic waste disposed of since the 1950s was recycled, whereas 12% was incinerated and the majority of the plastic waste remains in the landfills, dumpsites, and oceans worldwide [2]. Plastic waste has become one of the most concerning environmental issues as it has contributed to many environmental threats such as pollution of groundwater, marine litter climate change, and the release of hazardous substances. It was estimated that about 8 Mt of plastic has been entering the ocean annually and Vietnam is among the top five polluters globally [3].

China plays a very important role in the plastic waste stream worldwide. The country is not only the major plastic producer and consumer but was also the top plastic waste importer, with 56% of the total plastic scrap being imported into China [4,5].

Since China has tried to improve the quality of imported plastic waste with the ‘Green Fence Operation’ in 2013 and has permanently banned the import of certain types of household plastic waste starting in 2017 and coming into force in 2018, there has been a huge shift in the plastic waste import and export globally. With this restriction, the import has been decreased significantly for China and Hong Kong and plastic waste has shifted to other destinations within Southeast Asian countries. EU–27 countries were one of the main plastic waste exporters accounted for 31% of global plastic waste export, of which 85% is sent to China, where the plastic ban has had a major impact and created challenges for the redistribution and reduction in the export of plastic waste during 2017 and 2019 [6,7].

There is a high possibility that plastic waste import into Southeast Asian countries will increase and these countries will become the plastic world leaders in plastic waste imports [8] as between the years 2015 and 2018, 88 scrap plastic exporters diverted attention from China to other markets such as Turkey, with an increase of 1295%, Indonesia with an increase of 485%, Malaysia with an increase of 193%, and Vietnam with an increase of 111% [9].

This paper gives an overview of the plastic industry in Vietnam and how plastic waste is managed in this country. The recycling of plastic waste typically takes place in an informal context. To demonstrate this more in detail, two rural settlements–so-called craft villages—are taken as case studies. Technologies and processes for plastic recycling are described and related risks for human health and the environment are shown, as well as the potential for improvement of this situation.

## 2. Materials and Methods

The analysis of plastic waste management in Vietnam started from plastic production and waste management in the country. A closer view of the plastic industry in Vietnam gives insight into the dimensions of production and the demand for resources, which is partly covered by secondary materials. Subject to the analysis is the generation of plastic waste in the country and how the waste management system, in places, takes up this waste stream in terms of collection and processing. Both steps are mainly in the hands of the informal recycling sector. In a case study, the processing of plastic waste in an informal context is discussed, using data for two craft villages, where quantities, processes, and emissions are analyzed. As no samples were taken, the emissions from the recycling processes can be given only on a qualitative basis. However, by comparing the processes to other cases from the literature, the major impacts on health and the environment are identified. In the conclusions, recommendations are given on how to mitigate these adverse impacts.

## 3. Results

### 3.1. Plastic Manufacturing and Plastic Recycling in Vietnam

Vietnam is the 15th most populous country in the world with a total estimated population of 96.2 million in 2019. Approximately 65.6% of the population live in rural areas and 34.4% live in urban areas with the largest share in Hanoi, Ho Chi Minh City, Hai Phong, and Can Tho. [10]. According to the World Bank country classifications by income level, Vietnam is categorized as a low middle-income country with an average Gross National Income (GNI) per capita of USD 2,590 per year [11].

#### 3.1.1. Overview of the Plastic Industry in Vietnam

Vietnam’s plastic industry is one of the industries with relatively fast growth compared to the national economy in general. During the period from 2012 to 2017, Vietnam’s plastic industry grew on average 11.6% a year, faster than the world plastics industry’s 3.9% growth and faster than Vietnam’s average GDP growth of 6.2% over the same period [12] The plastic industry volume in 2017 was estimated at around USD 15 billion, equivalent to about 6.7% of Vietnam’s GDP in 2017 [12].

The majority of the product sectors are packaging (37%), household furniture (29%), construction (18%), and electronic appliances (29%) [13]. Therefore, Vietnam has become one of the top 20 global plastic products exporters and exports plastic products to more than 55 different countries [14].

In 2017, Vietnam’s plastic industry consumed about 5.9 Mt of virgin plastic materials, equivalent to a per capita plastic consumption rate of 63 kg/capita/year (kg/cap/year). The rate in 1990 was only 3.8 (kg/cap/year); thus, in the period from 1990 to 2017, the average plastic consumption per capita in Vietnam increased by 10.96% per year [12]. Similar quantities are reported by British Plastic Federation (55 kg/cap/year) and H.Böll Stiftung (41.3 kg/cap/year). Figure 1 shows the plastic consumption per capita in different regions.

However, Vietnam’s plastic industry is commonly focused on plastic processing and not fully active in inputting raw plastic materials sources to production activities. According to the Vietnam Plastic Association, the plastic industry in Vietnam needs approximately 3.5 Mt/year of the input raw materials such as polyethylene (PE), 30%; polypropylene (PP), 23%; polyethylene terephthalate (PET), 9%; and poly vinyl chloride (PVC), 5.7%, where the domestic production can only supply 0.9 Mt of the raw materials to the market [13]. Therefore, the plastic industry is highly dependent on the import of raw materials which continuously increase in quantity and value over the years. Even though there is a high potential for recovering the plastics raw materials from scrap, only 20% of the plastic wastes is recycled [16].

The volume of imported plastic materials of Vietnam’s plastic industry continued to grow in the period 2011–2017. The average growth in volume and value of imported raw materials during this period grew with an average of 11.5% and 7.8% per year, respectively [12]. This is an inevitable consequence of the fact that the upstream plastic industry did not develop fast enough to meet the rapidly growing demand of the downstream segment. The dependence of Vietnam’s plastic industry on imported materials has been forecast to continue in the future.

#### 3.1.2. Plastic Waste in Vietnam

Over the years, Asia has become the leading generator of plastic waste with 82 Mt in 2015, followed by Europe (31 Mt), and Northern America (29 Mt). Latin America, including the Caribbean, and Africa generated 19 Mt of plastic waste each while Oceania generated about 0.9 Mt [17].

Regarding the generation of plastic waste by country, China is the leading country with over 17 Mt/year, followed by India with more than 12 Mt/year. Vietnam was ranked 7th of the top country generators of plastic waste in the world, with over 1.8 Mt/year.

With an average share of 12% in municipal solid waste (MSW) (23.4 Mt, i.e., 242 kg/cap/year), the quantity of plastic waste in 2019 can be estimated at 29 kg/cap/year [12].

#### 3.1.3. Waste Management in Vietnam

The rapid population growth, increase in production, urbanization, industrialization, and economic development result in the intense extraction of natural resources and in increasing solid waste generation. According to the National Environmental Report, the total volume of MSW generated nationwide was about 16.2 Mt/year in 2011 [18]. By 2019, this figure was 23.4 Mt/year with an increase of 44% compared to 2011. The volume of MSW has increased significantly in regions with high urbanization, industrialization, and tourism such as Ho Chi Minh City (9400 t/day) and Hanoi (6500 t/day) [18,19,20,21]. MSW generated in urban areas is estimated at 13.0 Mt/year, accounting for 55% of the volume of MSW in the country, of which Ho Chi Minh City is the largest waste generator in the country, followed by Hanoi. Rural areas contribute another 10.4 Mt/year.

The per capita generation of MSW is highest in the northern midland and mountainous area with 1.20 kg/cap/day, followed by the north and south-central coast with 1.17 kg/cap/day; the lowest is in the Mekong Delta with 0.82 kg/cap/day. For rural areas, MSW generation in the Red River Delta is reported to be the highest with 0.52 kg/cap/day, followed by the north-central and central coast with 0.51 kg/cap/d; the lowest is in the northern midland and mountainous area with 0.29 kg/cap/day [18].

In terms of composition, Vietnam’s solid waste is characterized by high humidity (ranging from 65–95%), a high ash content of about 25–30% (dry weight), and low heat value (900–1100 Kcal/kg wet weight) [18]. The major waste composition is organic waste (67%) and other recyclable materials such as paper (6%), metal (6%), glass (4%), and plastic (12%) [22]. In household MSW, organic matter (food waste) is the largest constituent, with a decreasing trend. Since 1995, the composition of food waste accounted for a very high proportion (80–96%), but by 2017 this figure had decreased to about 50–70%, indicating that urban residents’ lifestyle changes are fast and convenient [19,23].

#### 3.1.4. Waste Collection and Treatment

In urban areas, the collection rate of MSW increased from 81% in 2010 to 85.5% in 2017 [18,24]. The collection and transportation of MSW are usually done by public enterprises. In recent years, with the policy of privatisation, private companies participate in the collection and transportation of MSW in urban areas. Funding for the collection and transportation of MSW is given by the state and stems from sanitation fees. In urban areas, municipal waste collection is undertaken by mobile garbage collectors, collection trucks, and through a container system [24,25,26].

In rural areas, the collection and transportation of MSW are mostly managed by cooperatives and self-managed collection teams. The rate and method of collection vary widely between localities. In many rural areas, due to inconvenience in transportation and lack of awareness of the population, waste is dumped into water bodies and on bare land without any management [25]. If MSW is collected, most of it dumped onto land, without any hygienic protection such as liners or leachate collection and covering, or it is exposed to uncontrolled burning.

Currently, there are 1322 MSW treatment facilities nationwide, including 381 solid waste incinerators, 37 composting plants, and 904 landfills, including a larger number of non-sanitary landfills (dump sites) [18].

From the total volume of MSW collected, about 71% is treated by landfilling, 16% is processed in composting plants, and 13% is treated by incineration. Out of the larger landfills, receiving more than 20,000 t/day, only 30% are classified as sanitary landfills as landfills with daily cover. Only 9% of the landfills are equipped with weights and 36% with bottom lining. Most landfills do not have compactors, a gas collection system, leachate treatment, or environmental monitoring [18].

The recycling of waste is mainly performed by the informal sector. Informal collectors purchase recyclable materials from households and industry. Wholesalers buy from informal collectors and industrial waste collectors; then sort, bale, and sell to processors [18,27]. In addition to the amount of recycled waste from the domestic market, a considerable amount is imported such as plastics (1.2 Mt/year) and paper (1.3 Mt/year).

The main limitation of the waste management system in Vietnam is the fact that a larger proportion of the waste generated is not collected, treated, or disposed of in a controlled manner. As a result, a number of environmental and health problems arise such as groundwater and soil pollution from leachate, methane emissions from the landfill, contaminated waterways, marine littering, air pollution from inadequate waste burning, and the spread of diseases.

#### 3.1.5. Management of Plastic Waste in Vietnam

The awareness of the majority of people of the sorting, collection, transportation, and treatment of MSW, especially plastic waste and plastic bags, is still limited. People are usually not aware of the harmful effects of plastic waste disposal to the environment and the ecosystem.

The collection and sorting of recyclable plastic waste from households is done by informal collectors. The collection rate for plastic waste is low, specifically for plastic bags, which are made of thin films and difficult to recycle. Producers of plastic packaging and other plastic products do not have a responsibility for the management of plastic waste.

Regarding the plastic recycling industry, most of the actors in the market are in the informal sector. The recycling of plastic waste still faces many difficulties, the reuse of waste is done through collection and transportation. It is then sent to the craft villages for recycling [25]. The recycling materials are mainly paper and plastics, which are processed manually and with outdated technology causing high emissions rates.

The topic of environmental pollution due to plastics in daily life has been integrated into local environmental protection and waste management plans [18]. Several cities and provinces have carried out public relations work, including information dissemination, and education to raise the awareness of the community about the harmful effects of non-biodegradable plastic bags. Movements such as “plastic-bag-free days”, “say no with plastic bags” plan and implement public events to combat plastic wastes. However, the size and number of the above-mentioned activities is still limited, and their effectiveness is not significant [18].

### 3.2. Plastic Recycling Practices in Craft Village

Accounting for more than 90% of activities, the informal sector is dominating plastic waste recycling in Vietnam. The activities of informal plastic waste recycling in Vietnam are carried out in craft villages, which play an important role in contributing to rural social–economic development and the industrialisation process. Beside recycling, typical activities in craft villages are silk processing, food production, or fine arts [28]. The craft villages are helping to alleviate poverty and hunger, create jobs, and increase income for people in the rural areas. However, despite many years of development, these activities are still bound to single households and recycling activities are performed on a small scales.

The basic production unit in a craft village is a household enterprise, which specializes in one or two recycling activities, e.g., waste collection, separation, shredding, or extrusion. The recycling enterprise has limited available working space but is tolerable for small-scale recycling operations. The equipment, machinery, and materials are set up next to or on the ground floor of the residential house, while the work force is mostly limited to the household members [29]. Households from the same craft village will mostly specialise in the same recycling activity. This allows them to cooperate, take over larger waste quantities, and benefit in general from economies of scale. As a result, a particular craft village will specialise in one type of recycling activity, whereas the neighbouring craft village will specialise in another, filling the gap in the recycling process. However, most of the craft villages’ recycling activities operate under unregulated and uncontrolled conditions, so that they are significant contributors to health hazards and pollution of the environment and the surrounding communities [28].

In 2018, the Vietnam Cleaner Production Centre (VNCPC) conducted a survey on two plastic recycling villages namely Phan Boi village and Minh Khai Village to analyze their recycling technologies and practices and their impact on the environment, human health, and the surrounding community [29].

Phan Boi and Minh Khai craft villages have specialised in different recycling activities but within the plastic waste recycling sector. While the majority of households in Minh Khai village process granulated plastic from waste scrap, the majority of households in Phan Boi village focus on waste plastic sorting, shredding, washing, and selling for further granulation processes [29].

In Phan Boi village, in 2018, about 130 households were working on plastic recycling activities. Of these 130 households, 81 (~62%) produced shredded plastic, 18 (~14%) prouced granulated plastic, and the remaining 31 (~24%) traded in waste materials [29]. As was also the case also in Minh Khai village, most of the activities were performed in the residential areas with limited working space. The input material in the village consisted of waste packaging, water pipes, pots, and other plastic items. On average, Phan Boi village has a processing capacity of between 100–200 t/day of unsorted input waste material. The main source of the material was the domestic waste plastic market. Because it usually arrived as a mixed waste stream, it was necessary to separate different recyclables and to sort and wash waste plastics before any further recycling steps [29]. About 85–88% of the input material is used for plastic recycling, while 5–10% consists of metal, paper, adhesive tapes, and others, which are then sold to the respective recyclers; 5–7% of input material remains unrecycled and it is often discarded in the dumpsite or on the roadsides of the villages [29].

In Minh Khai village, the majority of households were involved in agricultural production. Only a small number of households collected waste materials as a source of extra income. However, the trend changed, and by 2018 approximately 870 households were primarily working with waste plastics. Of these 870 households, approximately 260 households (~30%) were working on the plastic collection together with sorting and trading, 452 households (~52%) were involved with plastic granulation, 122 households (~14%) were dealing with films and other plastic products, and 35 households (~4%) were working with plastic shredding. Minh Khai village has a processing capacity of between 550–600 t/day of input material, which mostly consists of waste PE, PP, PVC, PS, HDPE, and LDPE plastics. It originates both from domestic and from international markets such as China, South Korea, Australia, and European countries. About 85–90% of input material is used for waste plastic recycling, 5–10% consists of other recyclables, and approx. 3–5% is disposed of in the proximity of the village in dumpsites and roadsides [29].

The main production outputs from Minh Khai village are PE and PP granules (90% purity), while the remaining outputs consist of other plastic products, such as plastic bags, trays, ropes, and buckets, and other recycling services, such as plastic shredding and trading in waste materials. The main recycling processes at Minh Khai village are sorting, cutting, shredding, washing and drying, extruding, and granulating [29]. The overview of the recycling processes in Minh Khai village is shown in Figure 2.

## 4. Discussion

### 4.1. Environmental Impacts

Waste plastics contain a wide range of different additives, which are mixed up with the polymer substrate in order to improve plastic characteristics such as durability, colour, and flammability. The most commonly used additives can be classified into the following categories with descending average concentration in the polymer substrate: plasticisers, flame retardants, stabilizers and antioxidants, slip agents, curing agents, biocides, colorants and pigments, and fillers [30].

During the informal recycling activities, waste plastics are exposed to mechanical, thermal, and chemical stresses, and a combination of these in uncontrolled conditions. For this reason, plastic additives tend to be released during the recycling process with the potential to cause adverse effects on health and the environment [29]. However, the plastic emissions during recycling activities are not inherently hazardous, but they should be regarded as potentially toxic substances (PoTSs) as defined by Hahladakis et al. [31]. Their toxicity arises from environmental conditions and stresses, exposure pathways, concentration, exposure duration, and other social-economic factors.

Since there was no possibility for us to take samples on-site, we have extrapolated potential emissions in Phan Boi and Minh Khai craft villages based on observed recycling activities and the average material composition of waste plastics. Furthermore, the described release pathways are cross referenced with the available data from literature for the same or similar informal emissions pathways (see Table 1).

The emissions from uncontrolled recycling activities and related concentration of hazardous substances have been analyzed for a number of locations, among others for flame retardants, heavy metals, and polycyclic aromatic hydrocarbons (PAH). As processes are similar, similar impacts can be expected for the craft villages considered here.

Tang et al. [53] investigated road dust in the area with intense mechanical recycling activities of waste plastics in Wen’an, north China. The analyzed dust samples showed between one and two orders of magnitude higher polybrominated diphenyl ether (PBDE) concentrations compared to outdoor and road dust samples from areas with no recycling activities. The commercial deca-BDE was the dominant type for approximately 85% of all detected PBDE. The analysis of heavy metals, showed average concentrations of arsenic (As), cadmium (Cd), chromium (Cr), copper (Cu), mercury (Hg), lead (Pb), antimony (Sb), and zinc (Zn) of 10.1, 0.495, 112, 54.7, 0.150, 71.8, 10.6, and 186 mg/kg, respectively.

Matuskukami et al. [54] studied organophosphorus flame retardants (o-PFRs), eight monomeric phosphorus flame retardants PFRs (m-PFRs), tetrabromobisphenol A (TBBPA), and polybrominated diphenyl ethers (PBDEs) in surface soils and river sediments near the recycling village Bui Dau, northern Vietnam. They found total PBDEs ranged from 67 to 9200 ng/g-dry in surface soils near the open storage area of recyclables, while the concentration near the open burning site of tris(methylphenyl) phosphate (TMPP) showed the highest concentration (2–190 ng/g-dry) of all measured flame retardants. They also concluded that the presence of o-PFRs is a good indicator of the substitution process of brominated FR to alternatives.

Polycyclic aromatic hydrocarbons (PAHs) are a large group of chemicals containing only carbon and hydrogen arranged in multiple aromatic rings and they are identified as carcinogenic organic compounds [55]. PAHs are mainly created during incomplete combustion and are very frequently detected near informal recycling sites. Since they are a side-effect of incomplete combustion, their presence can be attributed to the open burning of plastic.

Hoa et al. [56] investigated soil and sediment contamination by PAHs and methylated polycyclic aromatic hydrocarbons (MePAHs) in the area of an informal recycling site in northern Vietnam. They found an abundance of PAH and MePAHs contaminations (approximately 60% of all examined soil samples). The highest concentrations were found in workshop soil (median 2900; range 870–42,000 ng/g), followed by open burning soil (median 2400; range 840–4200 ng/g), paddy field soil (1200; range 530–6700 ng/g), and river sediment samples (median 750; range 370–2500 ng/g). However, the incremental lifetime cancer, the probability of developing cancer as the result of exposure to a specific carcinogen, of PAH-contaminated workshop soils was still within the acceptable levels of human health risk.

An investigation on plasticisers, PAHs, phthalic acid esters (PAEs), and bisphenol A (BPA) in the surface soil of informal recycling workshops in large Indian cities was carried by Chakraborty et al. [57]. They found that the average concentration of the 16 investigated PAHs (∑_16_PAHs) was 1259 ng/g, while the concentration of six PAEs (∑_6_PAEs) was 396 ng/g, and the average concentration of BPA was 140 ng/g. Furthermore, they concluded that the involvement of children and women in informal recycling might lead to their direct exposure and therefore risk of serious health problems.

### 4.2. Health Risks Arising from Informal Recycling

At the current stage, there is no health risk assessment of the plastic recycling activities for the workers and people who are living in the craft village, as there is no standard measurement for assessing the health risk potential. However, the exceeding of parameters and the contamination of the air and surface water can lead to health problems.

Informal recycling sites are known hotspots of dioxins and dioxin-related compounds (DRC), since they are formed during incomplete combustion at temperatures between 200 and 800 °C under the presence of BFRs and other halogenated FR as their chemical catalysts [58]. Such conditions are abundantly present during the open burning of waste plastics [59,60]. Dioxins are highly lipophilic compounds and insoluble in water, which triggers an easy transition from the environment to living organisms and subsequently humans [61]. Tue et al. [62] investigated the accumulation levels and profiles of DRC in breast milk samples from women living in the proximity of separate informal recycling sites. The results showed that women who are living in the proximity of informal recycling sites but are not directly involved, do not have significantly higher concentrations of World Health Organization toxicity equivalents (WHO-TEQ) compared to the control group of the Vietnamese background range (0.22–7.4 vs. 1.1–3.0 pg/g lipid). However, women directly involved in the informal recycling activities did have significantly higher concentrations of polychlorinated dibenzofurans PCDFs (13–15 pg/g lipid) and polybrominated dibenzofurans PBDFs (1.1–1.5 pg/g lipid) compared to the background range of 2.3–8.8 pg/g lipid and <1.1 pg/g lipid for PCDFs and PBDFs, respectively.

Cao et al. [63] assessed bioaccessibility and human health risk of Cu, As, Cd, Sb, and Pb in the soil near e-waste and waste plastic burning sites in Accra, Ghana using in vitro assay. The results of this study show elevated total concentrations of 211–20,400 mg/kg for Cu, 10–29 mg/kg for As, 7–29 mg/kg for Cd, 24–9450 mg/kg for Sb, and 24–10,800 mg/kg for Pb. The results for bioaccessibility-corrected human health risk assessment revealed noncarcinogenic risk for local inhabitants in half of the analyzed sites, while the carcinogenic risk was within an acceptable range. Sb together with Cu and Pb were identified to be one of the major metals of concern that contributed the most to the health risk.

## 5. Conclusions

The plastic industry in Vietnam is a strong and growing sector of the industry. The production—mainly for export—widely relies on imported raw materials, and secondary material (plastic waste) is partly used as a raw material input. Considering the management of municipal waste in Vietnam it becomes clear that today most of the waste is disposed of at landfills and dumpsites. Only a small part is sent for composting or recycling. Recycling mainly takes place in an informal context in craft villages. In craft villages, typically located in a rural area, residents have established additional economic activities, complementing agriculture. A number of craft villages have focused on recycling activities. However, due to a lack of state-of-the-art technology for recycling and a low degree of organization, the typical processes for plastic recycling in craft villages come along with health risks for workers and neighbours and a high potential of environmental pollution. This includes dust from sorting and shredding, wastewater from washing steps, and the uncontrolled disposal of residuals including uncontrolled burning. A major source of contamination is the emissions of VOC from the extrusion process.

To improve this situation, several measures can be taken, beginning with a more selective material intake to the recycling facilities and improvement in the sorting step in terms of sorting equipment and personal protection equipment. Wastewater from washing and shredding should undergo wastewater treatment before being released to the environment. For extrusion and granulation, the gaseous emissions should be reduced and at least collected and filtered. Finally, the disposal of residues should be organised in a better way; with partly hazardous materials open dumping is not an appropriate technology.

## Figures and Tables

**Figure 1 ijerph-18-04203-f001:**
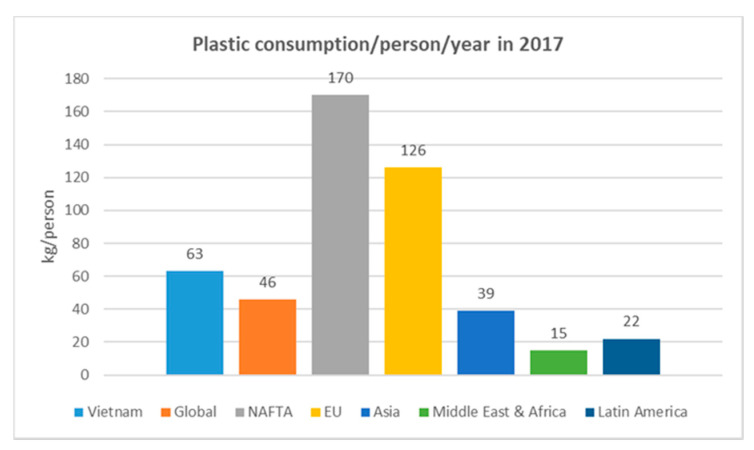
Per capita plastic consumption in 2017 (data from [15]).

**Figure 2 ijerph-18-04203-f002:**
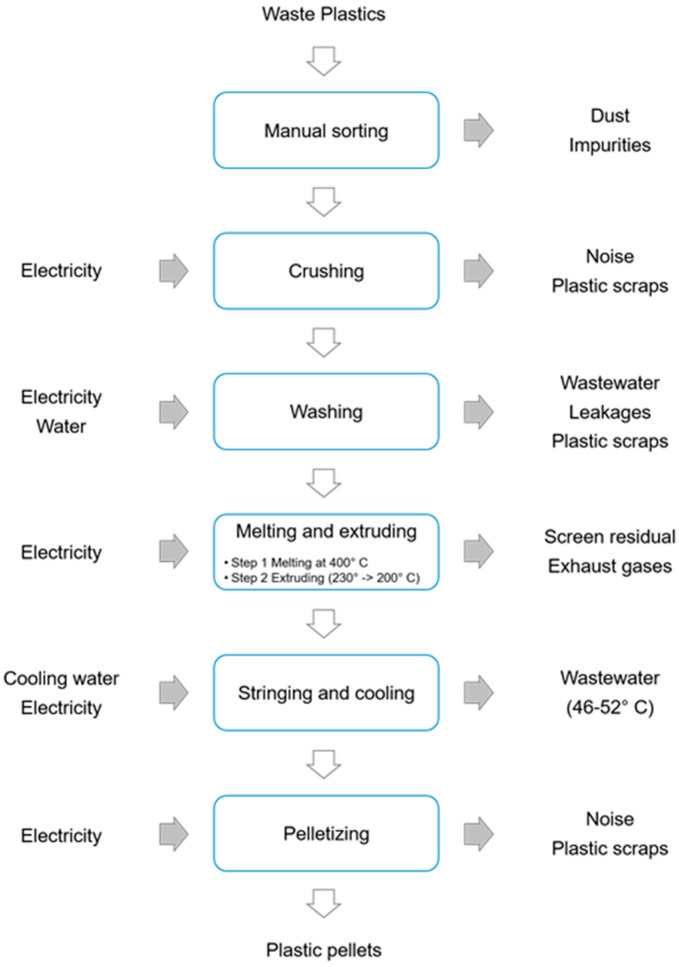
The plastic recycling process and related emissions at Minh Khai village [29].

**Table 1 ijerph-18-04203-t001:** Potential emissions resulting from specific informal waste plastic recycling at Phan Boi and Minh Khai craft villages activities with descriptions of their release pathways and corresponding scientific literature [29,32].

No	Process	Emission Potential Related to the Specific Recycling Activity	Description of Release Pathways	Release of Potentially Toxic Substances (PoTS)
1	Purchase and sorting of plastic waste	During collection, transportation, and sorting, the non-recyclable plastics and unintentionally lost recyclable plastics is piled-up along the roadside and the nearby landfill. There is approximately 65,000–100,000 t of plastic residues in the less strictly controlled landfill in Minh Khai.	Piled-up waste plastics in the landfill and along the roadside is exposed to weathering conditions such as sunlight (UV radiation), O_2_, and precipitation.	These conditions lead to increased brittleness of plastics, especially of polyethylene (PE), polypropylene (PP) plastic types, polyethylene terephthalate (PET), and polystyrene (PS). Consequently, following emissions of stabilisers, antioxidants, flame retardants, and micro plastics can be expected [33,34,35,36,37]
2	Washing of plastic	Small plastic pieces and dissolved plastic additives in wastewater of the washing process. About 6000–8000 m^3^ of wastewater per day is released without treatment	Washing water and washing supplements create a susceptible environment for the dissolution of plastic additives. Furthermore, the lack of any filtration system leads to microplastic emissions.	Migration and leaching characteristics of plastic additives include: plasticisers [38], brominated flame retardants [39,40], heavy metals [41,42], and curing agents [43]
3	Burning of sorted out plastics	When non-recyclable waste plastics is piled-up up to the extent that inhibits day-to-day activities, it is often set afire to reduce volume.	Open burning of plastics releases primary pollutants, i.e., substances contained in plastic material before burning, and secondary pollutants, i.e., substances catalyzed under thermal stress. If not directly inhaled in form of aerosols, the burning residues will end up in soil sediments, bioaccumulated in agricultural products, or in water bodies.	Open burning of plastics leads to emission of free radicals, heavy metals, polycyclic aromatic hydrocarbons (PAH), brominated flame retardants, and others [44,45].
4	Shredding	Prior to the extrusion process, the waste plastics is shredded to fine grains.	The shredders and crushers in Min Khai village do not possess any dust mitigation mechanisms. For this reason, the emissions settle in form of microplastic dust in the enclosed facilities or they are dispersed in the environment (soil sediments, bioaccumulation, water bodies) if the shredder is placed outside	The emissions of particulate matter from shredding process [46,47]
5	Plastic extrusion and granulation	Plastic extrusion is carried out on temperatures varying between 60–250 °C with a capacities between 500 and 750 kg/day depending on the plastic type	The plastic extrusion process produces air emissions and plastic residues on the sieve. The plastic residues are burned together with other plastics from the step 1	Emissions resulting from the thermal stress at temperature between 60 and 250 °C may include plasticisers [48], volatile organic compounds [49,50], ultra-fine particles [51], and flame retardants [52]

## Data Availability

All data in the paper refer to literature, which is cited.

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
