# Peer review of "Plastic Recycling Practices in Vietnam and Related Hazards for Health and the Environment"

_ijerph, 2021, doi:10.3390/ijerph18084203_

Round 1

Reviewer 1 Report

The paper is interesting if I can suggest one thing to the authors try to explain and summarize the data using more tables and graphs. Making the paper  more easily to follow and visually appealing to the reader is important.

Author Response

thank you for your comments, for the answers please see the document attached

Reviewer 2 Report

Plastic pollution is currently one of the biggest environmental concerns. Plastic waste has absorbed much concern in recent five years. The main problems are existing in some developing countries owing to inadequate supervision. This article presents how plastic waste handling in Vietnam, which has many implications from policy to technology. 

I know the authors have completed many countries' works on this topic from Asia to Africa. Could you compare them in this article?

thanks! 

Author Response

Thank your for the comments, for answers please note the document attached

Reviewer 3 Report

I thank the editor for the possibility to review the manuscript “Plastic recycling practices in Vietnam and related hazards for health and environment” (IJERPH-1146965). The theme is particularly important and the focus on local realities appears as timely and necessary. Moreover, Vietnam is a key geographical area for investigating several problems related to plastic waste (e.g., chemical, pollution, waste generation, marine litter) and solutions (e.g., collection, recycling, circular economy).

Unfortunately, the paper fails to address the topic satisfactorily and provide relevant knowledge for an academic readership.
A major problem that emerges immediately is the weak construction of the manuscript. A large portion of the manuscript is devoted to general and introductory issues (even when presented in the Result section) while the dynamics of the local realities are not thoroughly investigated. The root causes and local effects of waste are not scrutinised while the analysis on practices (what the paper is about considering the title) is short and remains at the surface.

More specifically, a reader may say that the (potentially) academically interesting part of the paper only starts on page 7, with section 3.2. All the previous text and information are rather general and too abundant for an academic paper of a certain quality and specificity. This does not mean that the first 6 pages are wrong but simply too broad, too general and not aligned to the objectives of the paper. This results in an extremely cumbersome portion of the manuscript and - in my view - deviate the paper from what could be a real contribution to an important topic. Just an example, lines 57-72 are about China and no link to the paper is visible.

The Result section provides (too) many pieces of information that are clearly of an introductory and compilatory nature. The plastic sector of Vietnam is considered with a little level of detail and the references are exceptionally limited if not absent. References 26 and 39 are not peer-reviewed while the vast academic literature on waste and plastic Vietnam is ignored. This lack of connection to the extant research is another major drawback of the manuscript. 

4.1.1 is entirely not about Vietnam and/or the areas of investigation. It is one entire page about addictive in plastic that seems – I say this with emphasis – to belong to another paper.

In 4.1.2. the sentence “Since we were not allowed to take samples on-site, potential exposure to the toxico-logical health impacts and environmental burden of informal recycling activities can be delineated by identifying employed recycling technology in Phan Boi and Minh Khai craft villages and compare these with the same or similar informal recycling activities from the literature” deprives the paper of a real academic significance. Moreover, using the term “from the literature” without any reference and explanation is not acceptable.

4.1.3. again, is not about Vietnam and/or the areas investigated.

It is evident that the authors have knowledge of the places and problems but they should consider rewriting the paper again. A paper framed differently, with a robust introduction and a few significant background data, a thorough analysis of problems and solution and a real (i.e., not interpolated) investigation has the potential to become good piece of research for such an important and relevant topic. In my view, the current manuscript is a lost opportunity. Despite my negative evaluation, which might be disappointing, I encourage the authors to reorganise the materials collected, conduct further research for relevant findings and entrench their research to the existing literature on Vietnam/waste/recycling.

Author Response

(The authors gave the same response as above.)

Reviewer 4 Report

Overall, I think that this a well written and interesting paper. Subject to some
minor revisions, I think that this paper should be published.

Corrections
* page 1 line 15 - remove "a waste stream of global dimensions"
* page 1 line 15 - replace "quick" with "rapid"
* page 1 line 17 - replace "plastic waste is taken by by a waste management system"
                    with "there are waste management systems"
* page 1 line 22 - replace "- so called craft villages-" with "(craft villages)"
* page 2 line 53 - remove "mainly"
* page 2 line 54 - add "," after "Asia"
* page 2 line 55 - remove "low to upper-middle income"
* page 2 line 56 - replace "high percentage of inadequate performance" with "an inadequate performance"
* page 2 line 76 - remove "+" in "+1295%" and "+485%"
* page 2 line 80 - replace "- so called craft villages-" with "(craft villages)"
* page 3 line 128 - should the best unit be "kg/year/person" rather that "kg/cap/year"?
* page 3 line 129 - assuming compound growth, it should 10.96%, not 10.6%
* page 3 Figure 1 - is there data available for 1990?
* page 4 line 139 - I think it would be neater to write "0.9Mt" rather than "900.000t"
* page 4 line 168 - remove "position"
* page 4 Table 1 - why is there no confidence interval for Vietnam, the one country we are interested in?
* page 5 line 182 - replace "210,000t" with "0.21Mt"
* page 5 line 191 - (1) be consistent - you have t/day and tonnes/day on the same line
                  - (2) maybe better to talk about Mt/year, rather that t/day
* page 5 line 216 and 217 - remove roman numerals (i) (ii) (iii)
                          - replace "collector" with "collection", "truck" with "trucks" and
                          "system" with "systems"
* page 5 line 218 - replace "garbage collectors collect solid waste in residential areas" with
                    "garbage collections, solid waste is collected in residential areas"
* page 5 line 220 - remove "Regarding"
* page 5 line 223 - replace "they are then collected" with "and then are collected"
* page 10 Table 2 - I would suggest to rotate the table, on a separate page if necessary.
* page 11 line 440 - replace the chemical symbols (As, Cd, Cr etc) with full names
                    (Arsenic, Cadmium, Chromium etc)
* page 11 line 441 - I assume "mg kg -1" is "mg kg^{-1}"? To be consistent with the rest of the paper,
                    you should use "mg/kg"
* page 11 line 460 - is the greater than sign (>) meant to be there, or is it mean to be a comma?
* page 11 line 468 - this is not my area of expertise, but what does the sum symbol mean in front
                    of 16PAHs and 6PAHs?

Author Response

(The authors gave the same response as above.)

Round 2

Reviewer 3 Report

No further comments